# The Effect of Soil Water Deficiency on Water Use Strategies and Response Mechanisms of *Glycyrrhiza uralensis* Fisch

**DOI:** 10.3390/plants11111464

**Published:** 2022-05-30

**Authors:** Kechen Song, Haiying Hu, Yingzhong Xie, Li Fu

**Affiliations:** 1College of Agriculture, Ningxia University, Yinchuan 750021, China; nxdxskc@163.com; 2Breeding Base for State Key Laboratory of Land Degradation and Ecological Restoration of North-Western China, Ningxia University, Yinchuan 750021, China; 3College of Politics and History, Ningxia Normal University, Yinchuan 750021, China; fuli39501793@163.com

**Keywords:** water stress, *Glycyrrhiza uralensis*, biomass allocation, water use efficiency, drought resistance

## Abstract

We aimed to investigate the water use strategies and the responses to water shortages in *Glycyrrhiza uralensis*, which is a dominant species in the desert steppe. Water stress gradients included control, mild, moderate, and severe. The time intervals were 15, 30, 45, and 60 d. Our study suggested that with the aggravation of water stress intensity, the total biomass of *Glycyrrhiza uralensis* gradually decreased and allometric growth was preferred to underground biomass accumulation. From 30 d and mild to moderate water stress, the water potential (WP) of leaves decreased considerably compared to the CK. The relative water content (EWC) decreased over time and had a narrow range of variation. Proline (PR) was continuously increased, then declined at 45–60 d under severe and more severe water stress. The δ^13^C values increased in all organs, showed roots > stems > leaves. The net photosynthetic rate (Pn) and transpiration rate (Tr) decreased to varying degrees. The instantaneous water use efficiency (WUEi) and limiting value of stomata (Ls) increased continuously at first and decreased under severe water stress. Meanwhile, severe water stress triggered the most significant changes in chloroplast and guard cell morphology. In summary, *Glycyrrhiza uralensis* could maintain water content and turgor pressure under water stress, promote root biomass accumulation, and improve water use efficiency, a water-conservation strategy indicating a mechanism both avoidable dehydration and tolerable drought.

## 1. Introduction

In the desert steppe, water is the most important factor limiting plant growth. Different plants adopt water use strategies to adapt to changes of water conditions [1]. A plant’s drought adaptability is closely related to their water use strategies [2,3]. Zhang et al. believes that drought resistance and the ability to use water effectively are essential mechanisms for revealing a plant’s drought resistance under long-term water stress conditions [4]. Therefore, the adaptation of plants to arid habitats requires the evolution of their water use strategies.

A previous study reported that plants could adjust the distribution of biomass among different organs to adapt to water stress, showing that the biomass of various organs of plants decreases due to water stress, while the root to shoot ratio (R/S) increases [5]. The biomass allocation strategy is also directly influenced by the ability of plants to adapt to their environment. Plants allocate resources to the most needed organs after environmental changes to effectively obtain scarce resources [6,7]. Plants obtain water by increasing root biomass when water is deficient. When soil moisture is sufficient, plants promote photosynthetic capacity by increasing the distribution ratio of above-ground biomass. However, above-ground biomass and underground biomass accumulation are not synchronized [8]. Therefore, plant organ biomass allocation results from a balance between reproduction and survival and a trade-off strategy for plants to adapt to their environment.

Precipitation can change Ci/Ca (intercellular CO_2_ concentration/atmospheric CO_2_ concentration) by changing the stomatal conductance of leaves, which then affects the δ^13^C value of plants [9]. The δ^13^C values of C_3_ plants in arid and desert areas ranged from −20‰ to −35‰ with decreasing precipitation. In extreme arid areas, the δ^13^C values ranged from −20‰ to −26‰, and precipitation has a significant negative correlation with the δ^13^C values [10]. It has been suggested that C_3_ plant tissues under water stress generally have higher δ^13^C values, while those under non-water stress have lower δ^13^C values [11], but most of the previous results were from field experiments. Liu Ying et al. studied the effects of different drought stress conditions on δ^13^C values of the C_4_ plant Leymus chinensis and found that the δ^13^C values were significantly positively correlated with water use efficiency (WUE) [12]. It is feasible to determine water use efficiency of *L. chinensis* by the δ^13^C value. However, some earlier studies suggested that δ^13^C values have growth stages and organ specificity when used to study WUE and biomass in plants [13].

*Glycyrrhiza uralensis* is the dominant species in the natural restoration of the desert steppe. It is not only a high-quality pasture and a Chinese herbal medicine, but it also has a higher ability to survive and compete, making it important in the study of grassland restoration. Hu found that *Glycyrrhiza uralensis* has the highest δ^13^C value and photosynthetic rate among four dominant species of the desert steppe, including *Glycyrrhiza uralensis*, *Lespedeza potaninii*, *Stipa breviflora*, and *Agropyron mongolicum*, under various precipitation conditions [14,15]. This research focuses on *Glycyrrhiza uralensis*. It uses a water control experiment to investigate the responses of biomass allocation, water use efficiency, and physiological and morphological characteristics to water stress to uncover the water use strategies and drought resistance mechanisms in *Glycyrrhiza uralensis*. It gives a scientific basis for choosing species to restore the grassland to its natural state.

## 2. Results

### 2.1. Biomass Allocation and the Root–Shoot Ratio of Glycyrrhiza uralensis under Water Stress

Under the same level of water stress, the total biomass of *Glycyrrhiza uralensis* increased first, then decreased; the highest biomass was obtained after 45 d of treatment. The root and leaf biomass accumulated quickly within 30–45 d, with root biomass at 45 d and 60 d significantly higher than at 15 and 30 d (*p* < 0.01). The root–shoot ratio (R/S) of *Glycyrrhiza uralensis* gradually increased over time, the R/S values under different water stress all showed 60 d > 45 d > 30 d > 15 d.

Under the same time, the total biomass of *Glycyrrhiza uralensis* decreased gradually as the degree of water stress increased, but the stem biomass remained unchanged. The R/S increased significantly (*p* < 0.05). The total biomass of the T4 treatment was the lowest, and the R/S of the T4 treatment was the highest at 15, 30, 45 and 60 d. These results indicated that drought stress resulted in more allocation of biomass to the roots and less allocation to the stems and leaves (Figure 1).

### 2.2. Growth Relationship between Above-Ground and Underground Biomass of Glycyrrhiza uralensis under Water Stress

Under different water stress treatments, there was an extremely significant correlation between the below-ground biomass (BGB) and the above-ground biomass (AGB) of *Glycyrrhiza uralensis* (*p* < 0.01), and the allometric growth relationship was biased toward under-ground biomass accumulation (Figure 2). Each treatment had a correlation growth index α greater than one. T2 had the highest α index (α = 1.9), the CK had the lowest, and T3 had the best fitting effect (R^2^ = 0.8592).

### 2.3. Effects of Water Stress on Water Potential of Glycyrrhiza uralensis Leaves

The leaf water potential (WP) of *Glycyrrhiza uralensis* decreased to varying degrees compared to the CK under water stress (Figure 3). With the increase in the water stress degree, WP remained unchanged at the early stages (15 d), decreased first, increased at the middle stages (30 and 45 d), and gradually increased at the late stages (60 d). Except for 15 d, the CK and T1 treatments had a significantly higher WP than T2, T3, and T4 treatments (*p* < 0.05).

The CK and T1 treatments fluctuated over time. The WP of the T2 and T3 treatment decreased rapidly at 30 d and then increased rapidly, the lowest at 30 d and the highest at 15 d. The WP at 15 d, 45 d, and 60 d was significantly higher than that at 30d, and the WP at 15 d was significantly higher than that at 45 d and 60 d (*p* < 0.05). The WP of the T4 treatment decreased rapidly and increased slowly, with the lowest at 45 d and the highest at 15 d. The WP at 15 d was significantly higher than that at 30 d, 45 d, and 60 d, and the WP at 30 d and 65 d was significantly higher than that at 45 d (*p* < 0.05).

### 2.4. Effects of Water Stress on Relative Water Content and Proline Content of Glycyrrhiza uralensis Leaves

The relative water content (RWC) of *Glycyrrhiza uralensis* leaves was significantly affected by both the degree of water stress and time passage (Table 1). The RWC continued to decrease slowly as the degree of water stress increased (*p* < 0.05), while proline content (PR) continued to increase significantly (*p* < 0.05). Compared with the CK, the RWC of T1, T2, T3, and T4 treatments decreased by 8.21%, 11.79%, 21.26%, 21.99%, and the PR of T1, T2, T3, and T4 treatments increased by 44.60%, 111.07%, 174.30%, and 246.91%, respectively.

The RWC at 30 and 45 d was significantly higher than that at 15 and 60 d (*p* < 0.05) as the stress time increased. The PR of *Glycyrrhiza uralensis* leaves increased at first, then decreased. At 30 d, the PR was significantly higher than at 15, 45, and 60 d; at 15 d, PR was significantly higher than at 45, and 60 d (*p* < 0.05).

### 2.5. Effects of Water Stress on δ^13^C Values in Different Organs of Glycyrrhiza uralensis

In different organs of *Glycyrrhiza uralensis*, the δ^13^C values were as follows: δ^13^C_root_ > δ^13^C_stem_ > δ^13^C_leave_, the δ^13^C values of different organs were significantly different (*p* < 0.05).

The δ^13^C values of all organs showed an upward trend as the degree of water stress increased. The δ^13^C_root_ of T2, T3, and T4 treatments was significantly higher than the CK and the T1 treatment. The δ^13^C_stem_ of the T4 treatment was significantly higher than the CK, T1, and T2 treatments, and the δ^13^C_leave_ was significantly higher than the CK, T1, T2, and T3 treatments (*p* < 0.05) (Figure 4).

### 2.6. Effects of Water Stress on the Gas Exchange Parameters of Glycyrrhiza uralensis

As demonstrated in Table 2, the Pn, Tr, Ci, and Gs of *Glycyrrhiza uralensis* leaves decreased with the increasing degree of water stress, the CK had the highest, and the CK and the T1 treatment were significantly higher than the T3 and T4 treatments (*p* < 0.05). The WUEi and Ls increased continuously at first, and then decreased significantly in the T4 treatment, with the WUEi of T1, T2, and T3 treatments being significantly higher than the CK and T4 treatment (*p* < 0.05).

### 2.7. Effects of Water Stress on Chloroplast and Stomatal Ultrastructure

Under normal water conditions, the chloroplasts of *G. uralensis* mesophyll cells were long and semi-circular, distributed close to the cell edge, with a complete structure and a clear membrane structure. Among them, the crenellations of thylakoids were tight and smooth, and the stromal lamellae were more evenly distributed and arranged, and there were a small number of mesophyll granules. The ultrastructure of chloroplasts changed to varying degrees as the degree of water stress increased (Figure 5a). The T4 treatment caused the chloroplast to become shorter and swollen to varying degrees, and the membrane structure of the chloroplasts gradually blurred. The number of osmiophilic granules increased and accumulated. Starch grains appeared, and their volume gradually increased. The number of granalamellae decreased, and the structure was fuzzy in the T4 treatment.

The guard cells in the leaves of *Glycyrrhiza uralensis* were found to have a typical reniform equithick wall, and the two guard cells were arranged symmetrically. Under normal water conditions, the thickness of the upper cell wall of guard cells was larger than that of the lower cell wall. The chloroplast, mitochondria, nuclear stomatal cavity, and other structures are clear and distinguishable, with normal morphology and more starch grains.

The guard cells became smaller as the degree of water stress increased, with uneven cell wall thickening (Figure 5b—T3, T4), protoplast shrinking volume (Figure 5b—T2, T3, T4), and starch grains gradually disintegrating. The stomatal cavity was shaped like a slender wine cup and severely deformed (Figure 5b–T4).

### 2.8. Analysis of Different Organ Biomass and Related Physiological Indexes of Glycyrrhiza uralensis

The biomasses of *Glycyrrhiza uralensis* organs were negatively correlated with R/S, δ^13^C_root_, and PR, and positively correlated with Tr and Gs (*p* < 0.05) (Figure 6). The δ^13^C values of each organ were positively correlated with the R/S and PR and negatively correlated with the leaf biomass, Pn, Tr, and Gs (*p* < 0.05). The R/S and PR were negatively correlated with the Pn, Ti, Ci, and Gs (*p* < 0.05). The RWC and WP were positively correlated with the Tr, Ci, and Gs and negatively correlated with the Ls (*p* < 0.05).

### 2.9. The Relationship between the Biomass of Different G.uralensis Organs and the Photosynthetic Physiological Indexes under Water Stress

Redundancy analysis revealed that RDA1 axis could explain 85.82% of the changes in organ biomass, which mainly reflected the changes in physiological regulatory factors. The proline content had the greatest influence on the changes of organ biomass, amounting to 54.3% of the explanations and 55.6% of the contribution rate (*p* < 0.01) (Figure 7). In addition, the Pn and δ^13^C roots were important factors affecting the changes in organ biomass, they accounted for 9.9% of the explanation, contributed 10.1%, and the δ^13^C_root_ had a significant influence (*p* < 0.05) (Table 3).

## 3. Discussion

### 3.1. Effects of Water Stress on Biomass Allocation of Glycyrrhiza uralensis

In response to water stress, changes in plant biomass and its distribution ratios to different organs reflect its adaptation methods and abilities to the ecological environment [16]. Meanwhile, there are stable allometric relationships between plant organ biomass and plant metabolism, this change in growth relationships may be the mechanism of drought tolerance in plants [17]. In the present study, the above-ground biomass and total biomass of *Glycyrrhiza uralensis* decrease to varying degrees, and the R/S significantly increased under severe water stress. Thus, the proportion of underground biomass was increased by slowing down the growth of the above-ground part, falling leaves, and other adaptive characteristics. This may be because plants often allocate more biomass to the ground to cope with extreme drought [18]. Under water deficit conditions, *Glycyrrhiza uralensis* increases the available soil water by increasing the root–shoot ratio and thereby boosting drought tolerance [19]. We also found a correlation between underground biomass and above-ground biomass under different water stresses (*p* < 0.05). The allometric growth content was greater than one, which further indicates that *Glycyrrhiza uralensis* allocates more resources to enhance the ability of water acquisition by the roots.

### 3.2. δ^13^C Value Composition of Glycyrrhiza uralensis Organs under Water Stress

In this study, the δ^13^C values of the roots of *Glycyrrhiza uralensis* were significantly higher than the leaves and stems (*p* < 0.01), and the δ^13^C values of different organs were as follows: δ^13^C_roots_ > δ^13^C _stems_ > δ^13^C _leaves_. Similarly, Gao also confirmed that the ^13^C abundance of fine roots, thick roots, and thick branches was significantly higher than the leaves [20]. This may be because roots accumulate ^13^C more easily than leaves and stems. Photosynthetic organs usually contain a low δ^13^C value, and most of the stems of herbaceous plants are also green photosynthetic organs. Therefore, the isotopic fractionation metabolism of the stems of herbaceous plants is closer to the leaves than that of wood and roots [21,22]. This indicates that the ^13^C fractionation produced by the leaves and stems was smaller than the roots. The main reason for plants ^13^C fractionation may be the heterogeneity of the chemical composition of their organs, and organs with high cellulose content are more likely to enrich ^13^C [23].

With water stress intensifies, plant species always have improved their δ^13^C values, and WUE will increase with increasing δ^13^C values [11,12]. The *Glycyrrhiza uralensis* in this study also conforms to this feature. However, we found that the leaf δ^13^C value of seedling was significantly lower than that of uncultivated *Glycyrrhiza uralensis* [14,15], indicating that wild plants or cultivated perennials have higher leaf WUE than young plants. This indicates that δ^13^C values of plants have organ and growth stages specificity under water stress [13].

### 3.3. Responses of Water Physiological Characteristics of Glycyrrhiza uralensis to Water Stress

Li suggested two types of drought-resistant plants: the first one has a high-water potential, a delayed dehydration, and drought tolerance, and the second one has a low water potential, tolerant dehydration, and drought tolerance [24]. The first type of plant delays the occurrence of dehydration through limited water loss, maintaining water absorption capacity. The second type of plant tolerates dehydration by increasing the water absorption capacity, maintaining turgor pressure, and reducing water loss. Yang and Song argued that under water stress, plants with less evident changes in water potential have a stronger ability to maintain normal turgor pressure [25,26]. They can maintain a specific amount of water and nutrient transport capacity. In this study, the leaf WP of *Glycyrrhiza uralensis* showed a downward trend as the intensity of water stress increased, dropping sharply under moderate water stress at the earlier times and increasing under severe stress at the late stages of stress. This demonstrates that *Glycyrrhiza uralensis* could regulate osmotic potential by decreasing the leaf WP, enhancing water absorption, resisting tissue dehydration, and being sensitive to drought-resistance.

Meanwhile, *Glycyrrhiza uralensis* accumulated PR in a large amount under water stress at the earlier times and decreased at the late stages of stress. Although *Glycyrrhiza uralensis* decreased the RWC with the intensity of water stress but stabilized it throughout all stress times. This showed that *Glycyrrhiza uralensis* maintained a certain turgor pressure to resist dehydration through the coordination of different physiological activities [27]. Based on the classification proposed by Li, we categorized this plant as close to the second type of drought-resistance [24].

### 3.4. Photosynthetic Physiological Properties of Glycyrrhiza uralensis Respond to Water Stress

The photosynthetic capacity of plant mesophyll cells is determined to some extent by the mesophyll cell’s ultrastructural characteristics [28]. In this study, water stress had a significant effect on the ultrastructure of epidermal cells of *Glycyrrhiza uralensis* leaves. Under severe water stress, the grana of chloroplasts underwent changes such as bending, swelling, and disarrangement, and osmophilic grains (lipid bodies) significantly increased and aggregated. Some chloroplast envelopes ruptured or even disintegrated, and inclusions flowed out. These results are consistent with the findings of various previous studies [29,30]. The guard cells of *Glycyrrhiza uralensis* were severely deformed, the cell wall was inhomogeneously thickened, the organelles and solutes decreased, and the cells were damaged. At this point, osmotic substances in guard cells were expelled or consumed, leading to water outflow and stomatal closure [31]. It was further confirmed in the present study that *Glycyrrhiza uralensis* was sensitive to water stress and regulated stomata by osmotic substance content and morphological changes.

In general, the response mechanism of photosynthesis is different under different water stress conditions. the decrease of Pn and Ci under mild water stress is the result of stomatal restriction. Under severe water stress, the photosynthetic apparatus and the photosynthetic enzyme system of plants were destroyed, and the non-stomatal restriction was the main reason for the decrease of Pn. Due to the decrease of the photosynthetic enzyme activity of mesophyll cells, the discrimination of δ^13^C was weakened, which resulted in the increase of the δ^13^C value, but the WUEi decreased [20,32]; our data also confirmed this feature of *Glycyrrhiza uralensis*. Therefore, under mild water stress, the Pn and Ci of plants decrease, and their absorption of ^12^CO_2_ is relatively reduced, but their δ^13^C value and water use efficiency increase due to stomatal restriction [33,34]. In this study, the change patterns of Pn, Tr, Gs, and Ci of *Glycyrrhiza uralensis* belonged to stomatal restriction under the T1-T3 treatments. Under severe water stress, the decreased range of Pn was higher than Tr, which led to the WUEi and Ls decreasing significantly, but the δ^13^C value increased. Thus, a non-stomatal restriction may play a dominant role under severe water stress when combined with changes in morphological and water physiological characteristics.

### 3.5. Water Use Strategy and Adaptation Mechanism of Glycyrrhiza uralensis in Response to Water Stress

Many studies have shown that plants adopt a survival adaptation strategy in arid habitats by improving the WUE and coordinating related physiological and metabolic functions to maintain the plant’s maximum water absorption capacity or minimum water loss to maintain survival [35,36]. Gulías found that neither *F. arundinacea* nor *D. glomerata* cultivars showed a trade-off between high WUE and biomass production, indicating that these plants have both the characteristics of efficient water use and productivity retention [2], whereas the other types of plants usually sacrifice above-ground biomass for efficient water use [37]. Here, we found that the δ^13^C values of *Glycyrrhiza uralensis* organs were significantly positively correlated with PR and root–shoot ratio and significantly negatively correlated with leaf biomass and gas exchange parameters. The change in water use efficiency was consistent with the change of root biomass and its reaction direction. Redundancy analysis further verified these results; physiological water regulation of *Glycyrrhiza uralensis* was the main component in the response vector to water stress. So, we concluded that *Glycyrrhiza uralensis* maintains water content and turgor pressure, promotes root biomass accumulation by increasing the root WUEi, resists tissue dehydration with a higher RWC, and maintains a specific photosynthetic yield, which is a water conservation strategy.

As the above analysis showed, in response to water stress, there was a significant trade-off between the biomass accumulation and water use efficiency of *Glycyrrhiza uralensis*. *Glycyrrhiza uralensis* responded positively to drought stress by increasing the root-to–shoot ratio and WUE while reducing biomass accumulation, regulating water status to match soil water supply capacity by osmosis and stomatal regulation, and maintaining relative humidity at a stable RWC, so as to reduce the risk of hydraulic imbalance. However, in order to ensure the safety of the plant hydraulic system and maintain the viability of the plant, the adaptation mode will change continuously or transitively with the change of drought degree [38].

## 4. Materials and Methods

### 4.1. Seed Collection and Nursery

The mature seeds of *Glycyrrhiza uralensis* were collected from the desert steppe natural restoration area in Gaoshawo, Yanchi County, Ningxia. Seedlings were grown in plugs in April and transplanted when the seedlings of *Glycyrrhiza uralensis* grew to more than 5 cm.

### 4.2. Soil Treatment and Its Physical and Chemical Properties

The desert soil was collected from the field sampling site at Gaoshawo, Yanchi County, and was transported to the agricultural experiment base (greenhouse) of the College of Agriculture, Ningxia University, China. The temperature inside the greenhouse was 28 °C/16 °C (day/night), with natural light. The soil was packed into self-made PVC plastic tubes with a diameter of 20 cm and a height of 40 cm after three days of sun exposure and screening out weeds and stones. Each tube contained 7.5 kg of soil, and the bottom of the tube was sealed with hard gauze. The field water holding capacity (FC) was 20.44 ± 2.74%, the volume weight was 1.45 ± 0.06 g·cm^−3^, and the saturation moisture capacity was 28.58%, close to the field sampling site. Bao’s method was used to investigate the physical and chemical properties of soil [39]: soil organic matter was 5.80 g·kg^−1^, total nitrogen was 0.38 g·kg^−1^, total phosphorus was 0.22 g·kg^−1^, total kaliun was 19.19 g·kg^−1^, alkali-hydrolyzable nitrogen was 27.33 mg·kg^−1^, rapidly available phosphorus was 5.27 mg·kg^−1^, available potassium was 113.31 mg·kg^−1^, total salt was 0.058 g·kg^−1^, and pH values were 9.45.

### 4.3. Design of Experiment

The experiment adopted a randomized block design with two factors (water gradient × times), and the water gradient treatment was divided into 5 gradients, each with three blocks.

Control (CK) for normal water supply, maintaining field water capacity between 70–80%. Treatment 1 (T1) was mild water stress, and field water capacity was between 60–70%. Treatment 2 (T2) was moderate water stress, and the field water capacity was 40–60%. Treatment 3 (T3) was relatively severe water stress, and field water capacity was 30–40%. Treatment 4 (T4) was severe water stress, and field water capacity was 20–30%.

After the seedlings were transplanted and new roots grew, five plants with the same growth were kept in each pot. The soil moisture was balanced regularly (2–3 d), and the field water capacity was maintained around 70–80%. After 15 d of growth, the moisture was being controlled. A TDR soil moisture meter (Mini Trase with soil-moisture TDR Technology, USA) was used to measure soil volumetric moisture content in each pot daily to ensure the soil’s water content in each treatment reaches the set level. The weight of each pot was measured every two days with an electronic scale. According to the soil moisture content and water consumption in each pot, the additional water contents were calculated to ensure the set range. The surface of each pot was covered with polyethylene plastic particles to prevent water evaporation.

### 4.4. Test Items

(1) Plant biomass of different organs

During different periods (15, 30, 45, and 60 d) of water stress, *Glycyrrhiza uralensis* was harvested and returned to the laboratory. The roots, stems, and leaves were put into paper bags and dried in an oven at 65 °C for 48 h, the dry weight of these plant components was measured. Each treatment was repeated five times.
Root to shoot ratio (R/S) = aboveground biomass/underground biomass.
Aboveground biomass = dry weight of stem biomass (SB, g) + dry weight of leaf biomass (LB, g).(1)
Underground biomass = roots biomass dry weight (RB, g).(2)

The linear correlation between the above-ground and underground biomass of *Glycyrrhiza uralensis* was analyzed using the correlation growth relationship model (*Y* = *β X^α^*). In the formula, Y is the underground biomass (BGB), β is the scaling constant, X is the above-ground biomass (AGB), α is allometric growth index, α = 1 is the isometric growth relationship, and α ≠ 1 is the allometric growth relationship. After the logarithmic transformation of biomass data, its power function is converted to the form of log = logβ + αlogX, where α is the slope of linear regression, and logβ is the intercept of linear regression.

(2) Relative leaf water content (RWC), free proline content (PR), and the leaf water potentials (WP)

During different periods (15, 30, 45, and 60 d) of water stress, fully expanded, healthy leaves were collected, while old leaves and new leaves were avoided. After sampling, a part of each plant was quickly determined for the fresh weight (FW) of the leaves, put in a container with distilled water (plastic bag), and let to stand for more than 12 h in a refrigerator at 4 °C. Then the absorbent paper was used to absorb water on the surface of the leaves and measure the weight of the leaves, which were the saturated weight (TW) of the leaves. Then the leaves were dried in an oven at 65 °C for 48 h, and the dry weight (DW) of the leaves was measured.
RWC (%) = (FW − DW)/(TW − DW) × 100.

The free proline content was determined by the acidic-ninhydrin method.

The situ determination method was used to measure the water potential of plant leaves in the early morning, and the Wescor psychrometer (PSYPRO water potential measurement system) was used to suite the C-52 sample room. During the determination, the plant leaves were clamped in the plant in-situ leaf chamber, balanced for 1 h, and the leaf chamber was completely sealed with plasticine. The leaves of the tested plant completely covered the measurement chamber. After 1 h, the probe was connected to the PSYPRO host to measure the leaf water potential.

(3) Carbon stable isotope composition (δ^13^C) of different organs.

Plant leaves of each treatment at the end of water stress were sampled, 15 leaves of five plants (3 leaves each plant) as one sample. Fully expanded, healthy, and southern leaves were selected from the sample plants. The plant leaves were dried at 60 °C for 72 h and then ground into powder and passed through a 100-mesh sieve for carbon isotope analyses, which were conducted at the Huake Jingxin Stable Isotope Laboratory of Tsinghua University, Shenzhen, China. The ^13^C/^12^C isotope composition was determined from 1-mg samples with a DELTAV Advantage isotopic ratio mass spectrometer (Thermo Fisher Scientific, Inc., USA). After high temperature combustion in the element analyzer, the samples generated CO_2_. The mass spectrometer detected the ^13^C and ^12^C ratio of CO_2_ and compared them with the international standard (Pee Dee Belnite or PDB, a shell fossil in the ocean, with a ^13^C content of 1.124%), then calculated the δ^13^C value of the samples. The δ^13^C values were expressed in parts per thousand (‰) and expressed as follows:$$\delta^{13}C\;\left(‰\right)=\left(R_{\mathrm{sample}}/R_{\mathrm{standard}}-1\right)\times 1000,$$
where *R* is the molar ratio of the heavy to light isotopes in the sample relative to the appropriate standards. The Pee Dee Belemnite carbonate was used as the standard for C. The accuracies of analyses were < ±0.1‰.

(4) The gas exchange parameters.

The gas exchange parameters of flag leaves were measured between 9:00 and 11:00 using a potable photosynthesis analyzer (LI-6400 by Li-Cor, USA) at the end stage of water stress. The gas exchange measurements were the average of three readings within 15 s. Three leaves lied on the equal node in each pot were determined, averaging each parameter of three leaves.
Instantaneous water use efficiency (WUEi) = Pn/Tr,
stomatal limit value (Ls) = (1 − Ci/Ca) × 100,
where Pn is net photosynthetic rate, Tr is transpiration rate, Ci is intercellular CO_2_ concentration, Ca is atmospheric CO_2_ concentration

(5) Observations of the leaf ultrastructure.

After 60 d of water stress treatment, the upper functional leaves of each treatment (five plants with basically the same growth status) were sampled at 8:00–9:00 in the morning under sunny weather. Six plants with basically the same growth status were selected from each treatment, and two upper functional leaves were taken from each plant. Afterwards, it was refrigerated at 4 °C and brought back to the laboratory. A sharp double-sided blade was used to crosscut the rectangular pieces of approximately 1 cm × 0.5 cm along the middle of the main veins of the leaves. The pieces were immediately fixed at 4 °C for 3 h with a 4% glutaraldehyde prefixative solution, then washed with 0.1 m sodium dimethyl arsenate three times, and the washing solution was replaced at an interval of 2 h. Then it was fixed with a 1% post-fixative solution of osmium at 4 °C for 2 h, washed twice with 0.1 m sodium dimethyl arsenate at an interval of 15 min, and dehydrated in gradient alcohol at room temperature, permeated and embedded with epoxy resin Epon812, and polymerized for 48–72 h in an incubator at 60 °C.

The embedded plant leaves were made into semi-thin slices with a thickness of 2–4 μm on leica UC-6 ultra-thin slicing machine. The test material was positioned under an optical microscope to determine the structural parts to be observed. After localization, leica UC-6 ultrathin slicing machine was used to slice the ultrathin slices with a thickness of 70–80 nm. The ultrathin slices were observed and photographed by JEM-2100HC transmission electron microscope after dual staining with uranium dioxy acetate and lead citrate.

### 4.5. Statistical Analyses

The original data were collated and displayed in Excel 2010. IBM SPSS Statistics 23 software was used to perform single or double factorial analysis of variance. Duncan’s method for multiple comparisons was adopted when there were significant differences. Correlation analysis (CA) and redundancy analysis (RDA) was performed on the data by R software and Origin 2018.

## 5. Conclusions

Under water stress, more biomass was allocated to the underground part of *Glycyrrhiza uralensis*, resulting in a significant increase in the R/S ratio. The biomass of above-ground and underground parts showed allelic growth biased to the accumulation of underground biomass. The δ^13^C values in all organs of *Glycyrrhiza uralensis* increased, the WP and PR of leaves significantly responded to mild and moderate water stress from 30 d and the RWC was exhibited a lower range of change. *Glycyrrhiza uralensis* exhibited sensitive responses to water shortages and maintained a certain turgor pressure to resist dehydration, which is a water conservation strategy. Meanwhile, the WUEi and Ls increased continuously and decreased significantly under severe water stress. The most significant morphological changes in chloroplast and guard cells began at T3 treatment. So, a non-stomatal restriction may play a dominant role under severe water stress.

We summarized that *Glycyrrhiza uralensis* could maintain water content and turgor pressure under water stress, promote root biomass accumulation, and improve water use efficiency, which was a water-conservation strategy showing a mechanism for both drought tolerance and avoidance.

## Figures and Tables

**Figure 1 plants-11-01464-f001:**
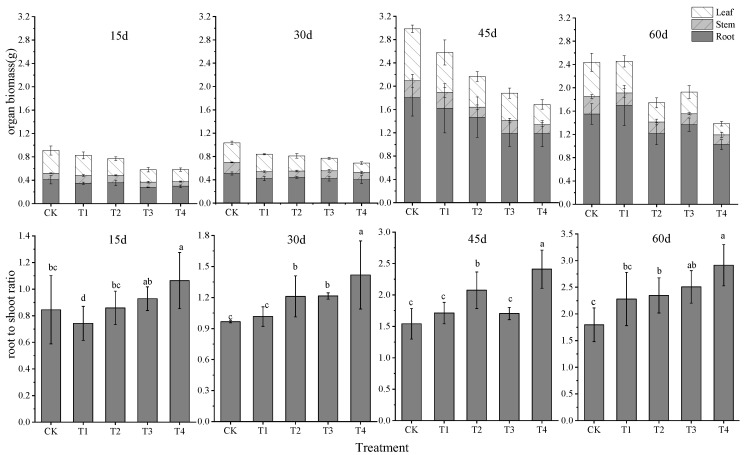
The biomass distribution in organs of *Glycyrrhiza uralensis* under different water stress treatments. Different lowercase letters indicate significant differences among treatments at 5% level.

**Figure 2 plants-11-01464-f002:**
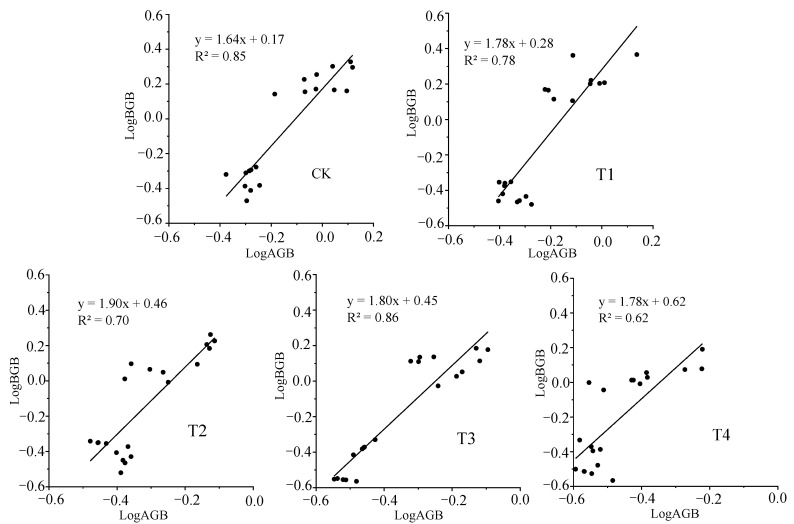
Growth relationship between above-ground and underground biomass of *Glycyrrhiza uralensis* in different water stress treatments. AGB and BGB represent the above-ground biomass and below-ground biomass. The biomass data are logarithmic transformed, its power function is converted to the form of log = logβ + αlogX, where α is the slope of linear regression, and logβ is the intercept of linear regression.

**Figure 3 plants-11-01464-f003:**
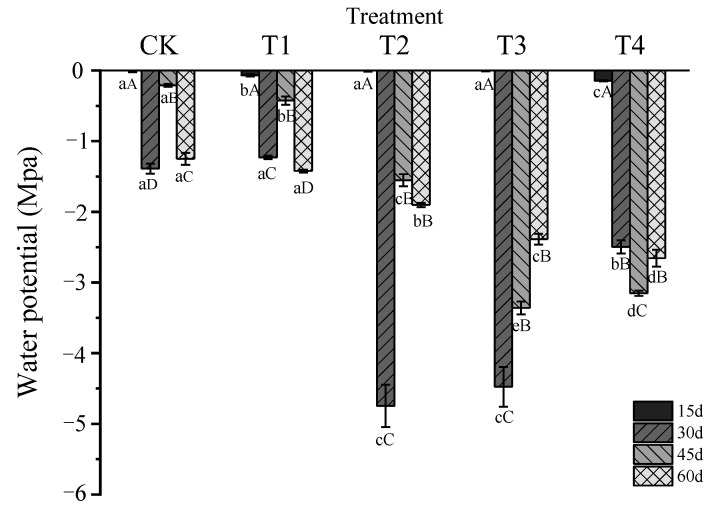
Changes of water potential in leaves of *Glycyrrhiza uralensis* under different water stress treatments. Different lowercase letters indicate a significant difference among treatments at a 5% level. Different capital letters indicate a significant difference at a 5% level between different times.

**Figure 4 plants-11-01464-f004:**
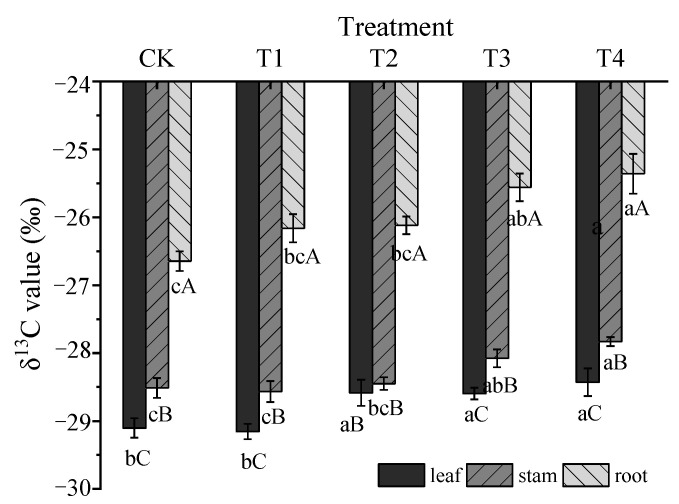
The composition of δ^13^C value in organs of *Glycyrrhiza uralensis* under different water stress conditions. Different lowercase letters indicate a significant difference among treatments at the 5% level. Different capital letters indicate a significant difference at a 5% level between different organs.

**Figure 5 plants-11-01464-f005:**
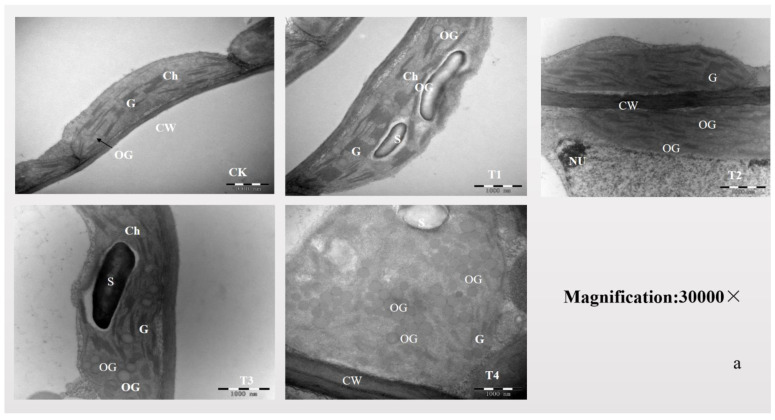
(**a**,**b**): Effects of water stress on chloroplast and stomatal ultrastructures of *Glycyrrhiza uralensis*. Ch, Mi, G, S, OG, P, CW, and N represent the chloroplast, mitochondrion, granalamellae, starch grain, osmiophilic granule, plastoglobulis, cell wall, and nucleus, respectively.

**Figure 6 plants-11-01464-f006:**
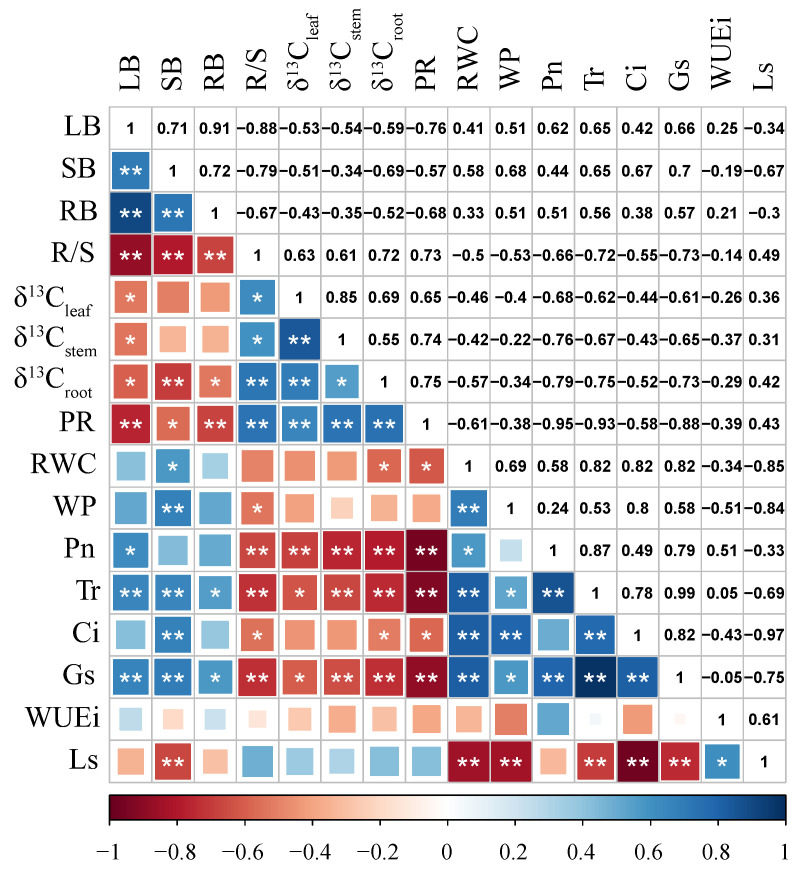
Correlation coefficient between different organ biomasses and related physiological indexes of *Glycyrrhiza uralensis*. δ^13^C_leaf_, δ^13^C_stem,_ and δ^13^C_root_ represent the δ^13^C values of leaf, stem, and root; LB, SB, and RB represent the leaf biomass, stem biomass, and root biomass; R/S, PR, RWC, WP, Pn, Tr, Ci, Gs, WUEi, and Ls represent the root to shoot ratio, proline, relative water content, water potential, net photosynthetic rate, transpiration rate, intercellular CO_2_ concentration, stomatal conductance, instantaneous water use efficiency, and limiting value of stomata, respectively. * and ** indicate significant correlation at 0.05 and 0.01 levels, respectively.

**Figure 7 plants-11-01464-f007:**
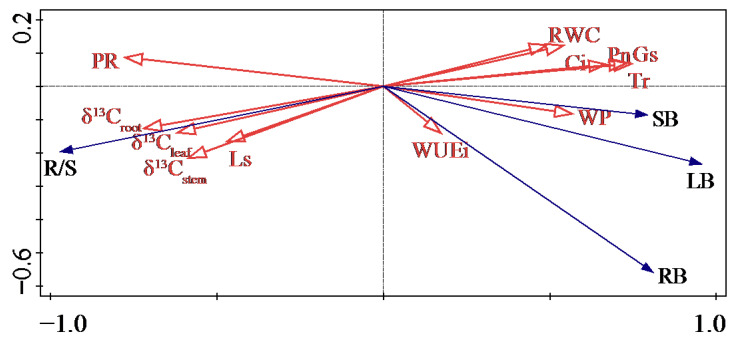
Results by redundancy analysis between biomass and physiological indexes of *Glycyrrhiza uralensis*. δ^13^C_leaf_, δ^13^C_stem_, and δ^13^C_root_ represent the δ^13^C values of leaf, stem, and root; LB, SB, and RB represent the leaf biomass, stem biomass, and root biomass; R/S, PR, RWC, WP, Pn, Tr, Ci, Gs, WUEi and Ls represent the root to shoot ratio, proline, relative water content, water potential, net photosynthetic rate, transpiration rate, intercellular CO_2_ concentration, stomatal conductance, instantaneous water use efficiency, and limiting value of stomata, respectively.

**Table 1 plants-11-01464-t001:** The effects of different water stress treatments on RWC and proline content of *Glycyrrhiza uralensis*. Note: Different lowercase letters indicate a significant difference among treatments or times at a 5% level.

Treatment	Relative Water Content (RWC)	Proline Content (PR)
CK	85.48 ± 1.59a	70.46 ± 9.90e
T1	78.46 ± 1.95b	101.89 ± 15.39d
T2	72.40 ± 2.17c	148.72 ± 18.24c
T3	67.31 ± 2.25d	193.28 ± 19.29b
T4	66.68 ± 5.75d	244.43 ± 13.54a
15 d	67.81 ± 3.20c	176.51 ± 16.90b
30 d	78.84 ± 2.24a	201.02 ± 21.84a
45 d	78.25 ± 1.55a	119.35 ± 17.16c
60 d	71.36 ± 1.54b	110.14 ± 14.60c

**Table 2 plants-11-01464-t002:** Effects of water stress on gas exchange parameters of *Glycyrrhiza uralensis*. Different lowercase letters indicate a significant difference at a 5% level between different times.

Index	CK	T1	T2	T3	T4
Net photosynthetic rate (Pn)	19.74 ± 0.77a	17.61 ± 0.37b	16.84 ± 0.65b	10.14 ± 0.92c	3.62 ± 0.24d
Transpiration rate (Tr)	17.16 ± 0.56a	11.39 ± 0.20b	10.42 ± 0.44b	6.10 ± 0.46c	4.27 ± 0.45d
Intercellular CO_2_ concentration (Ci)	356.90 ± 2.41a	334.15 ± 2.31b	319.93 ± 2.05c	293.90 ± 5.39d	315.64 ± 5.79c
Stomatal conductance (Gs)	1.07 ± 0.07a	0.50 ± 0.23b	0.39 ± 0.02c	0.18 ± 0.01d	0.10 ± 0.01d
Instantaneous water use efficiency (Wuei)	0.34 ± 0.02b	0.67 ± 0.03a	0.84 ± 0.04a	1.70 ± 0.14a	3.18 ± 0.33c
Limiting value of stomata (Ls)	1.15 ± 0.04d	1.55 ± 0.02c	1.63 ± 0.09b	1.66 ± 0.06a	0.87 ± 0.08bc

**Table 3 plants-11-01464-t003:** Results by redundancy analysis ordination with the first two axes and Monte Carlo permutation test.

Name	Explains %	Contribution %	Pseudo-F	*p*
Proline (PR)	54.3	55.6	15.4	0.002
Net photosynthetic rate (Pn)	9.9	10.1	3.3	0.072
δ^13^C values of root (δ^13^C_root_)	9.9	10.1	4.2	0.022
Stomatal conductance (Gs)	6.2	6.4	3.2	0.062
Water potential (WP)	2.8	2.8	1.5	0.25
Instantaneous water use efficiency (WUEi)	1.5	1.5	0.8	0.506
Transpiration rate (Tr)	0.8	0.8	0.4	0.72
δ^13^C values of stem (δ^13^C_stem_)	1.5	1.6	0.7	0.546
δ^13^C values of leaf (δ^13^C_leaf_)	5	5.1	3.1	0.084
Limiting value of stomata (Ls)	1.7	1.7	1.1	0.386
Intercellular CO_2_ concentration (Ci)	0.9	0.9	0.5	0.648
Relative water content (RWC)	3.2	3.3	2.7	0.14
Statistic	Axis 1	Axis 2	Axis 3	Axis 4
Eigenvalues	0.8582	0.0754	0.0426	0.0003
Explained variation (cumulative)	85.82	93.36	97.62	97.65
Pseudo-canonical correlation	0.9959	0.9214	0.9671	0.9563

## Data Availability

Not applicable.

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
