# Peer review of "The Effect of Soil Water Deficiency on Water Use Strategies and Response Mechanisms of Glycyrrhiza uralensis Fisch"

_plants, 2022, doi:10.3390/plants11111464_

Round 1

Reviewer 1 Report

Dear authors,
thanks for the interesting research. However, I believe that the presentation of the manuscript can and should be improved.

Abbreviations in the abstract make it difficult to understand the meaning of the study. It would be helpful to add a list of abbreviations at the end of the manuscript.

Species names should be written in italics.

It is necessary to expand the description of the results, in the current version they are presented very briefly and may not be clear to a wide circle of readers.

Figure 6: please write variables in one style: delta13… or d13…, R.S or R/S

It is necessary to expand the description of the methods, to clarify the data on the conditions for plants growing.

In the PDF file, I highlighted the most incomprehensible places.

Reviewer 2 Report

The paper "The effect of soil water deficiency on water use strategies and 
response mechanisms of Glycyrrhiza uralensis Fisch" is well structured. It has high significant and novelty. The topic covered is absolutely interesting and it is presented in high quality. The aim of the work is very interesting, the introduction is complete and updated on the topic. The research design is appropriate. The methods are adequately described and the results are very clear. The conclusions are supported by the results and  arguments presented addressing the main question that are posed.

Author Response

Dear reviewer

Thank you for your letter.Your comments gave us a lot of encouragement and meant a lot to us.  In order to further improve the quality of the manuscript, we have made appropriate additions to the results and methods. 

We would love to thank you for allowing us to resubmit a revised copy of the manuscript and we highly appreciate your time and consideration.

Sincerely.

Kechen Song.

Reviewer 3 Report

- What is new in this manuscript?

- Title

Glycyrrhiza uralensis must be Italic. Also, must be italic in all sections of the manuscript.

- Abstract

The authors must write the full name before each abbreviation for each parameter.

- Introduction

The authors must write a full name for each abbreviation for the parameters for the first time.

Line 31 Change Zhang to Zhang et al

Line 38 Check again roots Crown ratio

Line 53 Ling Ying is not [12] in the references section? The authors must revise carefully the references in the text and must be similar to them in the references section.

- Results

Line 80 Correct 60 d. (Figure 1). to 60 d (Figure 1). The authors must revise typing errors in the full manuscript.

- Figures and Tables

- The authors must write a full name for each abbreviation in the legends of Figures and Tables.

- 15d, 30d, 45d…… must correct to 15 d, 20 d, 45 d……

Author Response

Dear reviewer, please find it attached!

Round 2

Reviewer 3 Report

Accept in present form 

Author Response

We apologize for our oversight and have deleted the first part of Discussion.
